# Experimental Study on the Interface Characteristics of Reinforced Crushed Rock Cushion Layer Based on Direct Shear Tests

**DOI:** 10.3390/ma16175858

**Published:** 2023-08-26

**Authors:** Liangliang Wang, Qianlong Zhu, Yan Jia, Hu Li

**Affiliations:** 1State Key Laboratory of Deep Geotechnical Mechanics and Underground Engineering, China University of Mining and Technology, Xuzhou 221116, China; 2School of Mechanics and Civil Engineering, China University of Mining and Technology, Xuzhou 221116, China; 3Xuzhou Highway Development Center, Xuzhou 221000, China

**Keywords:** geogrid, crushed rock cushion, direct shear test, strength characteristics, deformation characteristics

## Abstract

Through indoor large-scale direct shear tests, the interface characteristics of the crushed rock cushions layer reinforced with ParaLink geogrid were studied. The test results indicate that the shear strength of the crushed rock aggregate and the interface strength parameters have a non-linear relationship with the normal stress. The addition of the geogrid reduces the shear strength of the crushed rock aggregate and the interface strength parameters, which is mainly due to the relatively large size, small thickness, and high smoothness of the geogrid. The reinforced geogrid has a significant impact on the deformation and fragmentation characteristics of the crushed rock aggregate. It effectively suppresses the shear contraction and shear dilation effects of the crushed rock aggregate, reducing its peak compression and peak dilation angle. Furthermore, it inhibits the tendency of particle fragmentation in the crushed rock aggregate.

## 1. Introduction

Geogrid, a novel type of geosynthetic material, has been widely employed in various reinforced soil engineering projects. With the rapid development of railway construction in China, the use of geogrid to improve the uneven settlement caused by uneven foundation deformation has become one of the best choices to pursue economic and social benefits [1]. The reinforcing effect of geogrids is mainly manifested through the interaction between the geogrids and the reinforced soil. The interface friction characteristics between geogrids and soil are crucial for studying their interaction properties, revealing the reinforcement mechanisms, and determining the design parameters of reinforced soil structures [2,3,4]. Conducting experimental research on the interface characteristics of the crushed rock cushions layer reinforced with geogrid can provide valuable guidance for the design and application of the geogrid-reinforced crushed rock cushion layer.

Currently, the research on the interface characteristics of reinforced soil is mainly based on direct shear tests and pull-out tests. B. M. Bacas et al. [5] conducted large direct shear tests to investigate the frictional characteristics of three critical interfaces: geogrid-soil, geosynthetic membrane-soil, and geosynthetic membrane-geogrid. The results revealed that frictional behavior is the primary factor influencing the behavior of these interfaces. Ferreira et al. [6] conducted interface direct shear tests on residual soil-geosynthetic interfaces to study the influence of factors such as soil moisture content, soil density, and geosynthetic material type on the interface performance between soil and geosynthetic materials. Wang et al. [7] conducted direct shear tests to investigate the effects of geogrid and fill material types, compaction degree, moisture content, and shear rate on the interface characteristics of reinforced soil, and they established separate quantitative models to evaluate the strength of the transverse-geogrid-soil and longitudinal-geogrid-soil interfaces. Cen et al. [8] conducted a comprehensive study on the frictional characteristics and mechanical behavior of geosynthetic membrane-sand interfaces under monotonic and cyclic loading conditions at different intensities. They developed a non-linear segmented model to simulate the pre-peak stress-dependent stiffness and post-peak displacement associated with the monotonic softening behavior. Udomchai et al. [9] conducted large-scale direct shear tests to study the interface shear strength between Reclaimed Asphalt Pavement (RAP) and jute geogrid (RAP-geogrid). They also investigated the feasibility of jute geogrid as an environmentally friendly base material and proposed a generalized equation for predicting interface shear strength. Li et al. [10], through indoor direct shear and inclined shear tests, studied the interlayer bonding characteristics of geogrid-reinforced asphalt. They found that the direct interlayer shear strength of geogrid-reinforced asphalt is lower than that of unreinforced asphalt. The stiffness of the geogrid affects its inclined shear strength. The direct shear-stress versus shear-displacement curve exhibited an oscillation region, while the inclined shear curve was relatively smooth throughout the process.

At the same time, Moraci and Gioffrè [11] analyzed the role of transverse geogrids and the scale effect of the geogrids, as well as the interlocking effect between soil and geogrids, through large-scale soil-geogrid pull-out tests. Moraci and Cardile [12] conducted pull-out tests on coarse-grained soil using two types of geogrids. The study found that the interface friction coefficient between soil and geogrid was primarily associated with the expansion effect of the soil within a certain range on the interface. Based on DEM, Miao et al. [13] replicated pull-out tests of three-axis geogrids embedded in various ballast materials to demonstrate the influence of particle shape on the geogrid pull-out behavior. Abdi et al. [14], through pull-out tests on reinforced soil with different particle sizes and distributions, as well as under different normal pressures, concluded that the pull-out force increases with increasing soil particle size and non-uniformity, and with an increase in normal pressure. The soil particle size and distribution, as well as the confining pressure, significantly affect the shear band thickness, with particle size being a major influencing factor. Luo et al. [15] also conducted relevant research on the frictional characteristics of the interface between crushed stone soil mixtures and geogrids through large-scale direct shear and pull-out tests in the laboratory. Du et al. [16] studied the evolution process of the macro and micro characteristics of the aeolian sand-geogrid interface and the mechanical properties of the reinforcement through indoor pull-out tests and numerical simulation.

In summary, it is evident that experimental research on the interface characteristics of reinforced soil is an important aspect of studying the performance, failure modes, and reinforcement mechanisms of reinforced soil structures. Currently, the existing research on the interface characteristics of geogrids and reinforced structures primarily focuses on the behavior of the reinforcement-soil or gravel-soil mixture interfaces. Wang et al. [17] and Li et al. [18] conducted direct shear tests and cyclic shear tests, respectively, to study the strength characteristics and deformation characteristics of the crushed rock cushion layer. However, both studies did not consider the use of reinforced geogrids, and there is still limited experimental research on the interface characteristics between geogrids and the crushed rock cushion layer. The theoretical research on the interface characteristics, reinforcement mechanism, and calculation methods of the geogrid-reinforced crushed rock cushion layer in the railway subgrade is significantly lagging behind engineering practice, which hinders the rapid development of reinforcement projects. Therefore, conducting experimental research on the interface characteristics of the crushed rock reinforced geogrid in the cushion layer is of great significance for understanding the interface properties, revealing the reinforcement mechanism, optimizing engineering design, and enhancing engineering reliability.

## 2. A New Subgrade Structure Based on the Interlocking Interaction between Concrete Sloped Toe Wall and Mattress

In the existing engineering design, the concrete slope toe wall is usually set at the foot of the subgrade to restrain the lateral deformation of subgrade structure under the combined action of dead weight and vehicle load, and the geogrid-reinforced crushed rock mattress is set at the base to adjust the differential settlement of subgrade surface caused by uneven foundation deformation (Figure 1). In this traditional subgrade structure system, the concrete slope toe wall and geogrid-reinforced mattress have different functions, and each plays an independent role. This treatment measure can ensure the overall stability of the subgrade when the subgrade engineering characteristics are stable, and the bearing capacity is strong. However, in the weak foundation section, where the seasonal groundwater level fluctuates greatly, and the engineering characteristics of the foundation soil deteriorate obviously with the change of humidity, the mattress layer often causes problems such as the suspension or fracture of the grille and the slip of the foundation of the concrete slope toe wall due to excessive local subsidence, and the subgrade slope toe wall and the mattress layer cannot restrain each other and coordinate deformation.

In order to solve the problem that the concrete slope toe wall and the mattress cannot restrain each other and cooperate in deformation, the research group invented a roadbed structure based on the interlocking synergistic effect of the concrete slope toe wall and the mattress. As shown in Figure 2, the two ends of the grid are poured into the concrete slope toe wall to realize the constraint of the reinforcement on the lateral slip of the embankment wall, and the concrete slope toe wall can be used as the anchorage section of the reinforced body. Thus, the stability of the subgrade in the weak foundation section is guaranteed, the uneven deformation of the subgrade is reduced, and the subgrade disease is reduced. The high-strength ParaLink (the following text will be abbreviated as PL) geogrid is used in the new subgrade structure. The shear interface characteristics between the high-strength tensile grid and the crushed rock cushion correlate with the stress distribution and displacement field characteristics of the entire subgrade structure system under the action of train load. In this paper, a comprehensive study is carried out in combination with large-scale direct shear tests.

## 3. Experimental Research

### 3.1. Direct Shear Test Equipment

The direct shear testing apparatus used in this experiment is the DZJ-900 Large Direct Shear Apparatus, located in the State Key Laboratory for Geomechanics and Underground Engineering at China University of Mining and Technology. The apparatus consists of a shear box, horizontal shear device, normal pressure loading device, and data acquisition system. The schematic diagram of the entire experimental apparatus is shown in Figure 3, and the performance indicators are listed in Table 1.

The shear box is divided into upper and lower shear boxes, with internal dimensions of 300 × 300 × 150 mm and 300 × 300 × 150 mm, respectively. The horizontal shear device is equipped with an infinitely variable speed function for strain rate control. The shear device is controlled by a motor to achieve uniform shear displacement, applying horizontal load to the lower shear box. The normal pressure-loading device consists of an air compressor, a pressurizing cylinder, a connecting cylinder, a top pressure plate, and a reaction adjustment rod. It primarily uses a pressure regulator and pressure indicator to set the required normal load for the test. The data acquisition system mainly consists of vertical displacement sensors, horizontal displacement sensors, and horizontal load sensors. They, respectively, measure the vertical deformation of the soil during shearing, the horizontal deformation of the lower shear box, and the shear force on the upper shear box. The data collected by the sensors is transmitted to the software system, which automatically displays the variation of shear force and settlement displacement with shear displacement during the shearing process.

### 3.2. Materials

#### 3.2.1. Crushed Rock Aggregates

Extensive research has shown that in order to eliminate the influence of the size effect, the dimensions of the test apparatus should be at least 7 times larger than the particle size of the majority of the bulk material [19]. For the crushed rock cushion, the particle size typically ranges from 20 mm to 80 mm. Due to the limitations of instrument size, it is almost impossible to directly use the actual crushed rock aggregate used in engineering applications. In order to make the test results closer to real-world conditions, a similarity grading method was employed in the experiment, with a ratio of D/d = 4, using crushed rock aggregates with a particle size range of 5–20 mm. The crushed rock aggregate particles have distinct edges and corners. There are more regular block-shaped particles and fewer needle-shaped or flake-shaped particles. The particle size distribution curve and the composition of typical particles for the crushed rock aggregates were obtained through sieve analysis, as shown in Figure 4. The values of the coefficient of uniformity (C_u_) and the coefficient of curvature (C_c_) are 2.69 and 1, respectively, indicating that the crushed rock aggregate particle size distribution is relatively uniform.

#### 3.2.2. Geogrid

In the newly developed subgrade structure by our research team, the geogrid needs to be embedded within the concrete slope toe wall to balance the tensile forces generated during the non-uniform settlement deformation of the geogrid-reinforced cushion layer. Therefore, geogrids require high tensile strength and better durability. The geogrid used is a ParaLink geogrid produced by Maccaferri Company (Zola Predosa, Italy), and its technical specifications are shown in Table 2. It features an interlocking arrangement of transverse and longitudinal ribs, with a large mesh size, high tensile strength, and strong impact resistance. It is not easily punctured by sharp stones or crushed stones. The thickness of the nodes is greater than that of the ribs, and the node shape exhibits geometric symmetry. Please refer to Figure 5 for the specific shape of the geogrid.

### 3.3. Test Program

The termination shear strain selected for this direct shear test is 10%, corresponding to a final shear displacement of 30 mm. In order to ensure the compacted state of crushed rock aggregate, the void ratio is taken as the control target. The crushed rock aggregate was prepared with a controlled dry density of 1680 kg/m^3^ and a void ratio of 0.59. To facilitate loading and protect the test apparatus, a 20 mm space was reserved at the top of the upper shear box to accommodate a rigid pressure plate. Additionally, a 10 mm high-density polyethylene porous board was placed at the bottom of the lower shear box. Therefore, the total volume of the shear box is 0.3 m × 0.3 m × 0.27 m = 0.0243 m^3^. By converting the volume of the crushed rock aggregates to mass, the mass of each group of crushed specimens was calculated to be 40.82 kg.

In order to investigate the reinforcement effect of geogrid in the crushed rock cushion, direct shear tests were conducted on both unreinforced and ParaLink geogrid-reinforced crushed rock specimens. Previous studies [20] have shown that the shear rate has a limited influence on the shear characteristics of crushed rock aggregates. However, when the shear rate reached 3 mm/min, there were observed instabilities in the normal stress, and some stress–strain curves exhibited noticeable anomalies. Considering practical conditions and previous experiences, the direct shear tests on both unreinforced and ParaLink geogrid-reinforced crushed rock were conducted at shear rates of 0.8 mm/min under normal stresses of 25 kPa, 50 kPa, 100 kPa, and 200 kPa. To ensure the accuracy of the test results, two parallel tests were conducted on samples with abnormal data.

### 3.4. Sample Preparation

During the preparation of the direct shear specimens, the two types of crushed rock aggregates, 10–20 mm and 5–10 mm, are mixed in the specified proportions. After thorough mixing, the mixture is sequentially filled into the shear box. The crushed rock aggregates are filled in six layers based on their mass. After each layer is filled, it is manually leveled and compacted using a compaction instrument. Vertical stress of 400 kPa is then applied to consolidate the crushed rock aggregates by using the top pressure plate. The consolidation continues for 10 min after reaching a constant vertical displacement to ensure the crushed rock aggregates fill reaches the desired height. This process is repeated six times to fill the upper and lower shear boxes with compacted crushed rock aggregates.

When preparing the direct shear specimen of the geogrid reinforced with crushed rock aggregates, three layers of crushed rock aggregates are first filled according to their mass, and the lower shear box is completely filled with crushed rock aggregate. The geogrid is securely fixed on the outer left side of the upper shear box using epoxy resin, positioning the longitudinal ribs in the middle of the shear box. After arranging the geogrid, three layers of crushed rock aggregates are filled into the upper shear box. To ensure the strength of the geogrid embedded in the slope toe walls on both sides, during the construction process, the geogrid is laid along the longitudinal direction of the subgrade. Therefore, when conducting the direct shear test, cutting should be conducted in the transverse direction. Due to the large spacing between the longitudinal and transverse ribs of the geogrid used, during the test, only two transverse ribs and one longitudinal rib were cut. The specific preparation steps of the specimens are shown in Figure 6.

## 4. Results and Discussion

### 4.1. Strength Characteristics of the Crushed Rock Aggregates

#### 4.1.1. The Shear Force–Shear Displacement Curve of the Crushed Rock Aggregates

The shear force–shear displacement curves of the unreinforced and PL geogrid-reinforced crushed rock aggregates obtained from the direct shear tests are shown in Figure 7. From Figure 7, it can be observed that the shear force–shear displacement curve of the unreinforced crushed rock aggregates exhibits a peak and shows a typical strain-softening behavior. The shear force–shear displacement curve of the PL geogrid-reinforced crushed rock aggregate is relatively stable, demonstrating a weak strain-softening behavior. The occurrence of strain-softening becomes more pronounced as the normal stress increases.

According to the pattern observed in the curves, the shear force–shear displacement curves of the unreinforced and PL geogrid-reinforced crushed rock aggregates can be divided into three stages:(1)Linear growth stage: In the initial stage, the shear force–shear displacement curve of the unreinforced crushed rock aggregate exhibits a nearly linear rapid growth phase. As the normal pressure increases, the shear force of the crushed rock aggregates increases at a faster rate with shear displacement. The slope of the initial straight line also becomes steeper. Conversely, for the PL geogrid-reinforced crushed rock aggregates, as the normal pressure increases, the growth rate of the shear force becomes slower, and the slope of the initial straight line becomes lower;(2)Fluctuation Growth Stage: With the increase in shear displacement, the shear force gradually and slowly fluctuates and grows until it reaches the peak shear force. During the shearing process, the shear “jumping” phenomenon occurs due to the uneven gradation of the crushed rock aggregates, shear displacement, flipping, and fragmentation of the gravel particles. The shear force–shear displacement curve of the crushed rock aggregate is significantly influenced by the normal stress. The higher the normal stress, the greater the shear force of the crushed rock aggregates and the larger the shear displacement required to reach the peak shear force (except for 50 kPa). This is mainly because, at lower normal stresses, the rolling and repositioning of aggregate particles are relatively easy, allowing them to quickly reach a more stable state. On the other hand, higher normal stresses restrict the repositioning of crushed rock aggregate particles, requiring larger shear displacements to reach a stable state. In the direct shear process, the reinforced geogrid significantly influences the shear force of the crushed rock aggregate. The shear force of the PL geogrid-reinforced crushed rock aggregate is reduced compared to the shear force of the unreinforced crushed rock aggregate;(3)Shear Failure Stage: After reaching the peak shear force, the shear force-shear displacement curve gradually forms a continuous shear failure plane, and the “jumping” phenomenon in the shearing process continues. The shear force fluctuation decreases; the higher the normal stress, the more pronounced the downward trend. Compared to the PL geogrid-reinforced crushed rock aggregates, the unreinforced crushed rock aggregate shows a more pronounced downward trend.

#### 4.1.2. Evolution of Interface Shear Strength of Crushed Rock Aggregates

Shear strength is an important indicator reflecting the strength characteristics of crushed rock aggregates. According to the Code for Soil Test of Railway Engineering (TB10102-2010) [21], the peak or stable value of the shear stress–shear displacement curve is taken as the shear strength in direct shear tests on coarse-grained materials. In the absence of a clear peak, the shear displacement reaching 10% of the specimen diameter or length is taken as the failure criterion. In this test, peak or stable values were observed, so the shear stress corresponding to the peak or stable value was considered as the shear strength.

As crushed rock aggregate particles are cohesionless granular materials without cohesive forces, only particle interlocking, and frictional resistance exist. Representing the shear strength of crushed rock aggregates with a linear strength criterion implies that the intercept of the linear line representing cohesion, denoted as c, is not zero. Therefore, Mello et al. [22] proposed a non-linear strength criterion for crushed rock aggregates, which can be expressed as follows:(1)$$\tau_{f}=A\sigma_{n}^{b}$$

By employing the non-linear strength criterion shown in Equation (1), the experimental results of this study were described. It was found that the non-linear strength envelope could effectively describe the relationship between shear strength and normal stress (*R*^2^ > 0.98). The relationship between shear strength and normal stress for the unreinforced and PL geogrid-reinforced crushed rock aggregates obtained from the tests is shown in Figure 8. Equation (1) for the material of this research will be as follows:(2)$$\tau_{f_{0}}=12.497\sigma_{n}^{0.587}$$
(3)$$\tau_{f_{1}}=3.261\sigma_{n}^{0.816}$$

The initial stage of the curve exhibited significant nonlinearity, which gradually decreased with increasing normal stress. Consistent with the findings of Li and Zhao et al. [10,23], the shear strength of the PL geogrid-reinforced crushed rock aggregate is weakened compared to that of the unreinforced crushed rock aggregate. The shear strength reduction is particularly pronounced under low normal stresses, with a maximum reduction of 35.7% (σ = 50 kPa).

Studies have shown that the length, width, thickness of ribs, grid node thickness, relatively open area, and grid section of the geogrid all have an impact on the shear strength of the reinforced structure [23]. The cross-sectional thickness of the ribs and nodes, as well as the relatively open area of the geogrid, are positively correlated with the shear strength of the reinforced structure. On the other hand, the length and width of the ribs are negatively correlated with the shear strength of the reinforced structure. The interaction between the geogrid and the crushed rock aggregate mainly occurs through friction and interlocking effects. Due to the relatively small cross-sectional thickness of the PL geogrid-reinforced ribs and nodes, and the relatively large length and width of the ribs, as well as the smooth surface of the longitudinal ribs and fewer in quantity, the friction and interlocking forces at the interface between the geogrid and the crushed rock aggregate are smaller compared to the bite force, friction force, and crushing force between the crushed rock particles. As a result, the shear strength of the PL geogrid-reinforced crushed rock aggregate is lower than that of unreinforced crushed rock aggregate.

In the newly developed subgrade structure with a synergistic interaction between the slope toe wall and the geogrid-reinforced cushion layer by our research team, the PL geogrid-reinforced is cast inside the concrete slope toe wall. The slope toe wall serves as an anchor segment for the reinforced cushion layer to compensate for the weak layer created by the geogrid in the crushed rock. The primary purpose of reinforcing the geogrid is to ensure the stability of the railway subgrade and reduce uneven settlement deformation. When the geogrid achieves the desired tensile strength and effectively mitigates uneven settlement deformation, it remains highly applicable in this interlocking subgrade structure.

#### 4.1.3. Interface Direct Shear Parameters of the Crushed Rock Aggregates

(1) Peak friction angle

Considering that the shear strength envelope of the crushed rock aggregate is non-linear, and the strength parameters vary with normal stress. Many researchers [24,25] defined the peak friction angle ($\varphi_{p}$) as a shear strength indicator for the crushed rock aggregates, as shown in Equation (4).
(4)$$\varphi_{p}=\tan^{-1}(\frac{\tau_{p}}{\sigma_{n}})$$

In the equation, $\tau_{p}$ represent the shear strength (kPa), and $\sigma_{n}$ represents the normal stress at the interface (kPa).

The variation of the peak friction angle with normal stress is shown in Figure 9. It can be observed that the strength parameters of the crushed rock aggregates are dependent on normal stress. The peak friction angle representing the shear strength indicator of the crushed rock aggregates exhibits a non-linear decay in the form of a power function with increasing normal stress. The peak friction angle of the unreinforced 0-layer crushed rock aggregate decreased from 71.1° to 54.5°, while the peak friction angle of the PL geogrid-reinforced crushed rock aggregate decreased from 63.9° to 51.5°. As the normal stress increased, the rate of peak friction angle reduction gradually slowed down. When the geogrids were added, the peak friction angle of the crushed rock aggregate showed a certain degree of attenuation. The maximum attenuation of the peak friction angle is observed when the normal stress is 100 kPa, with a magnitude of 16.5%. Equation (4) for the material of this research will be as follows:(5)$$\varphi_{p_{0}}=112.62\sigma_{n}^{-0.133}$$
(6)$$\varphi_{p_{1}}=93.591\sigma_{n}^{-0.119}$$

Although the peak friction angle ($\varphi_{p}$) of the PL geogrid-reinforced crushed rock aggregates follows a power function decay with increasing normal stress, the fitting goodness is slightly lower, indicating that it may not accurately assess the friction characteristics at the geosynthetic-aggregate interface.

(2) Apparent friction coefficient

In order to better evaluate the friction characteristics at the geosynthetic-aggregate interface, many researchers [26,27] define the apparent friction coefficient (*f*^*^) at the interface as the ratio of shear stress to normal stress, namely:(7)$$f^{*}=\frac{\tau_{sg}}{\sigma_{n}}$$

In the equation, $\tau_{sg}$ represents the shear strength of the geosynthetic-aggregate interface (kPa). During the overall shearing process of the geosynthetic, the frictional resistance at the interface can be assumed to be uniformly distributed and balanced with the shear force. $\sigma_{n}$ represents the normal stress at the interface (kPa).

The apparent friction coefficient is a comprehensive strength parameter that reflects the frictional characteristics of the geosynthetic-soil interface. Unlike the aforementioned characteristic friction angle for crushed rock aggregates, the apparent friction coefficient is closely related to the type of geosynthetic, its roughness, and the properties of the fill material. The relationship between the apparent friction coefficient and the normal stress is shown in Figure 10. Equation (7) for the material of this research will be as follows:(8)$$f^{*}=4.85\sigma_{n}^{-0.269}$$

From Figure 10, it can be observed that the apparent friction coefficient decreases with increasing normal stress. As the normal stress increases, the rate of decrease in the apparent friction coefficient also decreases. After the normal stress reaches 100 kPa, the rate of decrease becomes very small. Surprisingly, under normal stress of 200 kPa, the apparent friction coefficient for the PL geogrid-reinforced crushed rock aggregate is greater than that under normal stress of 100 kPa. Therefore, in reinforced engineering, the appropriate apparent friction coefficient should be selected based on the actual stress conditions.

### 4.2. Deformation Characteristics of Crushed Rock Aggregates

#### 4.2.1. Vertical Displacement–Shear Displacement Curve of Crushed Rock Aggregates

Figure 11 shows the vertical displacement–shear displacement relationship curve of the unreinforced and PL geogrid-reinforced crushed rock aggregates during the shear process. As depicted in Figure 11, throughout the entire shear process, the shear strain–vertical displacement curves of the unreinforced and PL geogrid-reinforced crushed rock aggregates exhibit similar patterns, displaying a sequence of initial compression followed by dilation.

During the initial application of shear load, both the unreinforced and PL geogrid-reinforced crushed rock aggregates experience compression and exhibit a downward vertical displacement, indicating a compressive behavior. The compression stage of the crushed rock aggregate is relatively short and typically concludes within a shear displacement range of 0–5 mm. As the normal stress decreases, the amount of compression deformation decreases, leading to a less pronounced compressive behavior.

With the increasing shear displacement, the vertical displacement of unreinforced and PL geogrid-reinforced crushed rock aggregate gradually increases, resulting in an upward displacement, indicating the occurrence of dilation behavior. The dilatancy of crushed rock aggregate increases with the increase of shear displacement, and the dilatancy curves of unreinforced and PL geogrid-reinforced crushed rock aggregate fluctuate under different normal stresses, exhibiting certain “jumping” phenomena, but the overall performance is rapid first and stable. The greater the normal stress, the slower the dilatancy rate, the smaller the dilatancy amount, and the less obvious the dilatancy phenomenon. The larger the shear displacement required for dilatation. The inclusion of a geogrid reinforcement significantly influences the deformation characteristics of the crushed rock aggregates, effectively suppresses the dilation behavior of the crushed rock aggregate, and reduces the amount of dilation.

#### 4.2.2. Peak Compression and Peak Dilation Angle of Crushed Rock Aggregates

The shear-compression characteristics of crushed rock aggregates are often described by peak compression. To quantitatively characterize the shear dilation behavior of crushed rock aggregates, the concept of peak dilation angle ($\psi_{p}$), proposed by Asadzaden et al. [25], is introduced, as shown in Equation (9).
(9)$$\psi_{p}=\tan^{-1}{\left(d\delta_{v}/d\delta_{h}\right)}_{\max}$$

In the equation, $d\delta_{v}$ represents the vertical displacement increment, and $d\delta_{h}$ represents the shear displacement increment.

The relationship curves between the peak compression, peak dilation angle, and normal stress of unreinforced and PL geogrid-reinforced crushed rock aggregate are shown in Figure 12. From Figure 12a, it can be observed that peak compression of both unreinforced and PL geogrid-reinforced crushed rock aggregate linearly increases with increasing normal stress. When the geogrids were added, the peak compression of the crushed rock aggregate decreased significantly. The maximum attenuation amplitude reaches 47.8% (at 50 kPa). As the normal stress increases, the difference in maximum compressibility between the unreinforced and PL geogrid-reinforced crushed rock aggregate becomes larger. It increases from 0.074 mm at 25 kPa to 0.26 mm at 200 kPa. However, the attenuation amplitude reduces to 16.5%.

From Figure 12b, it can be observed that the peak dilation angle ($\psi_{p}$) of both unreinforced and PL geogrid-reinforced crushed rock aggregate decreases nonlinearly with increasing normal stress. As the normal stress increases, the rate of decay gradually decreases, and the peak dilation angle of the crushed rock aggregate tends to stabilize. When the geogrids were added, the peak dilation angle of the crushed rock aggregate experienced significant attenuation. As the normal stress increases, the difference in peak dilation angle between the unreinforced and PL geogrid-reinforced crushed rock aggregate becomes smaller, resulting in a smaller attenuation amplitude. The difference in peak dilation angle decreases from 16.79° at 25 kPa to 3.048° at 200 kPa, and the attenuation amplitude decreases from 40% to 20.7%.

The fitting formula of all deformation characteristics related to indexes of this material is shown as follows:(10)$$\delta_{h}{{}_{{}_{1}}}_{\left(\max\right)}=-0.007\sigma_{n}+0.123$$
(11)$$\delta_{h}{{}_{{}_{1}}}_{\left(\max\right)}=-0.008\sigma_{n}+0.028$$
(12)$$\psi_{p_{0}}=96.009\sigma_{n}{}^{-0.353}$$
(13)$$\psi_{p_{1}}=41.356\sigma_{n}{}^{-0.235}$$

In summary, geogrid reinforcement has a significant impact on the deformation characteristics of crushed rock aggregate, effectively restraining shear dilation and shear contraction. However, this effect weakens as the normal stress increases. Geogrid reinforcement can reduce the shear dilation and shear contraction of crushed rock aggregate. This is mainly due to the larger area of reinforcement provided by the PL geogrid, which creates an isolating effect. It prevents the free-rolling and sliding of crushed rock particles between the upper and lower shear boxes, and the volume expansion deformation caused by particle position adjustment becomes relatively smaller. Therefore, geogrid reinforcement modifies the deformation characteristics of the crushed rock aggregate. This result validates the feasibility of using PL geogrid as a material to improve uneven settlement in the foundation and provides design parameters for its application in the synergistic interaction of the slope toe wall and the geogrid-reinforced cushion layer in the interlocking subgrade structure.

### 4.3. Crushing Characteristics of Crushed Rock Aggregates

#### 4.3.1. Variation in Aggregates Composition

Crushed rock aggregates, granular materials, are widely used in construction projects such as rock-filled dams and road embankments. However, due to their angular and predominantly block-like particles, as well as the possibility of inherent fissures, particle crushing may occur under external loads. Particle crushing can easily lead to changes in the strength and exacerbation of deformation in granular materials, resulting in deformation and stability issues in construction projects. Wang et al. [28] discovered that particle crushing is the fundamental cause of the non-linear shear behavior of coarse-grained materials under high normal stress conditions. Therefore, conducting experimental studies on the crushing characteristics of crushed rock aggregate is of great significance for practical engineering.

In the aforementioned direct shear tests, the shear force–shear displacement relationship curves obtained for unreinforced and PL geogrid-reinforced crushed rock aggregate reveal significant fluctuations in shear force as shear displacement increases. The occurrence of shear “jumps” can also be observed. Based on extensive previous research [29], it can be inferred that these shear “jumps” primarily result from particle crushing, shear displacement, and flipping of particles during the shear process. Additionally, after the occurrence of distinct particle fragmentation sounds during shearing, an immediate decrease in shear force is observed.

To study the particle crushing characteristics of crushed rock aggregates, the changes in particle composition before and after the analytical tests were analyzed using the indicator method $\Delta W_{k}$ ($\Delta W_{k}$ = $W_{ki}$ − $W_{kf}$, $W_{ki}$ and $W_{kf}$ respectively represent the mass percentage of particles in the particle size interval *k* before and after the test). The particle size distribution and composition changes of the unreinforced and PL geogrid-reinforced crushed rock aggregates before and after the tests are shown in Figure 13.

From Figure 13, it can be observed that the content of each particle size group in the unreinforced and PL geogrid-reinforced crushed rock aggregates under different normal stresses has undergone certain changes. The particle size distribution of the crushed rock aggregates before and after the tests mainly shows an increase in the content of small and medium-sized particles and a decrease in the content of large-sized particles. The content of each particle size group is significantly affected by the normal stress. The changes in the content of each particle size group under different normal stresses are not the same.

#### 4.3.2. Relative Breakage Potential of Crushed Rock Aggregates

In the context of overall fragmentation, the commonly used metrics are the breakage potential (*B_p_*) and the relative breakage potential (*B_r_*), as proposed by Hardin [30]. As shown in Equation (14):(14)$$B_{r}=B_{t}/B_{p}$$

In this equation, on the particle size distribution curve, the area enclosed by the initial gradation, the 0.074 mm sieve size line, and the horizontal axis represent the breakage potential (*B_p_*). The area enclosed by the initial gradation, the post-test gradation, and the horizontal axis represent the total breakage potential (*B_t_*).

The variation of the relative breakage potential (*B_r_*) of the unreinforced and PL geogrid-reinforced crushed rock aggregate with normal stress is shown in Figure 14. It can be observed that the relative breakage potential (*B_r_*) of the crushed rock aggregate has a good linear relationship with the normal stress; the formula is shown as follows:(15)$$B_{r}{}_{{}_{0}}=10^{-4}\sigma_{n}+0.997$$
(16)$$B_{r}{}_{{}_{1}}=9\times 10^{-5}\sigma_{n}+1.005$$

The relative breakage potential (*B_r_*) increases with an increase in normal stress, indicating that higher normal stresses make the crushed rock aggregate particles more susceptible to breakage. Geogrid reinforcement also has a significant influence on the fragmentation characteristics of crushed rock aggregate. The relative breakage potential (_r_) of the crushed rock aggregate tends to decrease when the geogrids are added. It indicates that geogrid reinforcement can suppress the structural failure tendency of the crushed rock aggregate.

## 5. Conclusions

In this paper, the interfacial direct shear tests of unreinforced and PL geogrid-reinforced crushed rock aggregate under different normal stresses are carried out. Some of the conclusions of this study are summarized below:(a)Under different normal stresses, the shear force–shear displacement curves of crushed rock aggregates exhibit peak values, showing typical strain-softening behavior. The higher the normal stress, the greater the shear displacement required to reach the peak shear force. The shear force–shear displacement curve of crushed rock aggregates becomes more stable when the geogrids were added, exhibiting a weak strain-softening behavior;(b)The shear strength of crushed rock aggregate exhibits a non-linear increasing trend with increasing normal stress. The shear strength of the crushed rock aggregate decreased when the geogrids were added. The attenuation of shear strength is particularly significant at low normal stresses, with a maximum attenuation amplitude of 35.7%;(c)The peak friction angles ($\varphi_{p}$) of reinforced crushed rock aggregate decrease nonlinearly with an increase in normal stress. The apparent friction coefficient (*f*^*^) also decreases nonlinearly with an increase in normal stress. However, after the normal stress reaches 100 kPa, the variation of the apparent friction coefficient tends to stabilize;(d)The shear strain–vertical displacement curve of the crushed rock aggregate exhibits an evolution pattern of shear contraction followed by shear dilation. The crushed rock aggregate experiences significant attenuation in terms of peak compression and peak dilation angle ($\psi_{p}$) when the geogrids were added, with maximum attenuation amplitudes of 47.8% and 40%;(e)The particle size distribution of crushed rock aggregate before and after shearing mainly shows an increase in the content of small and medium-sized particles and a decrease in the content of large-sized particles. Geogrid reinforcement has a significant impact on the fragmentation characteristics of the crushed rock aggregate and reduces its relative breaking potential (*B_r_*).

## Figures and Tables

**Figure 1 materials-16-05858-f001:**
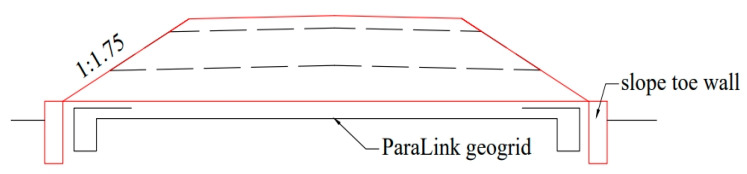
Slope toe wall and mattress of traditional railway subgrade structure.

**Figure 2 materials-16-05858-f002:**
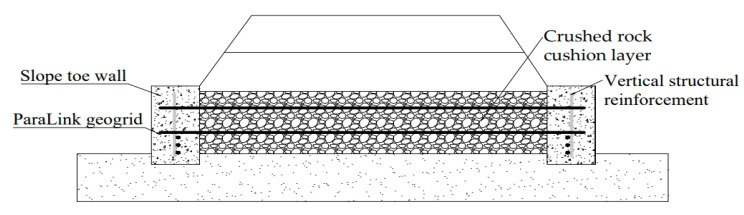
New subgrade structure based on the interlocking interaction between concrete slope toe wall and mattress.

**Figure 3 materials-16-05858-f003:**
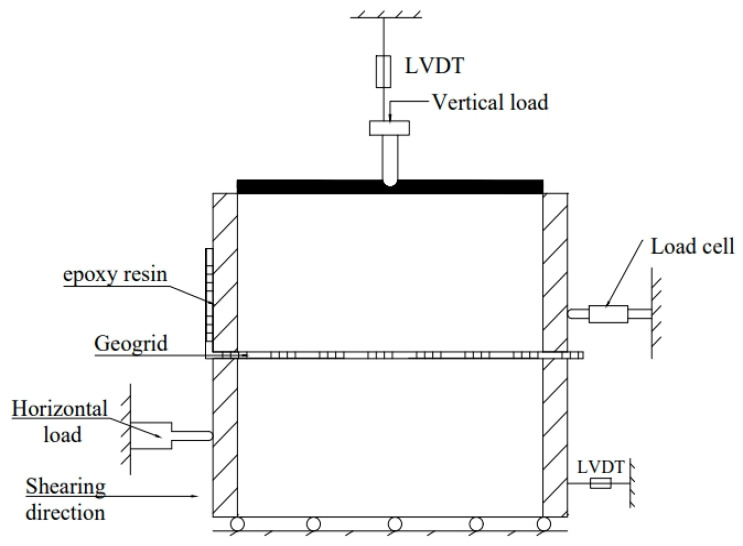
Schematic view of the large-scale direct shear apparatus.

**Figure 4 materials-16-05858-f004:**
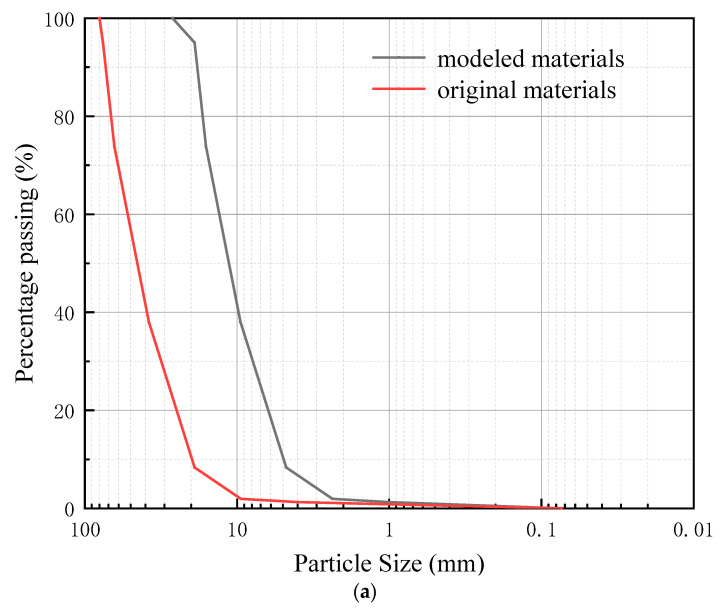
The crushed rock aggregates grading curve and typical particle composition. (**a**) Prototype and modeled crushed rock aggregates gradation curves; (**b**) typical particle composition of the crushed rock aggregates.

**Figure 5 materials-16-05858-f005:**
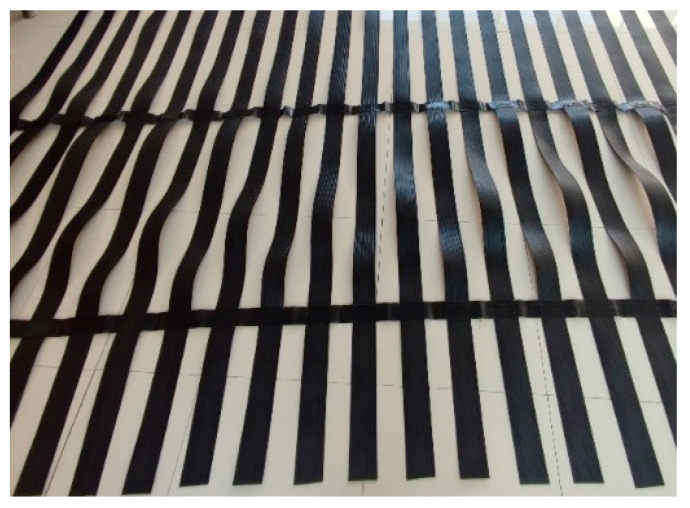
ParaLink Geogrid.

**Figure 6 materials-16-05858-f006:**
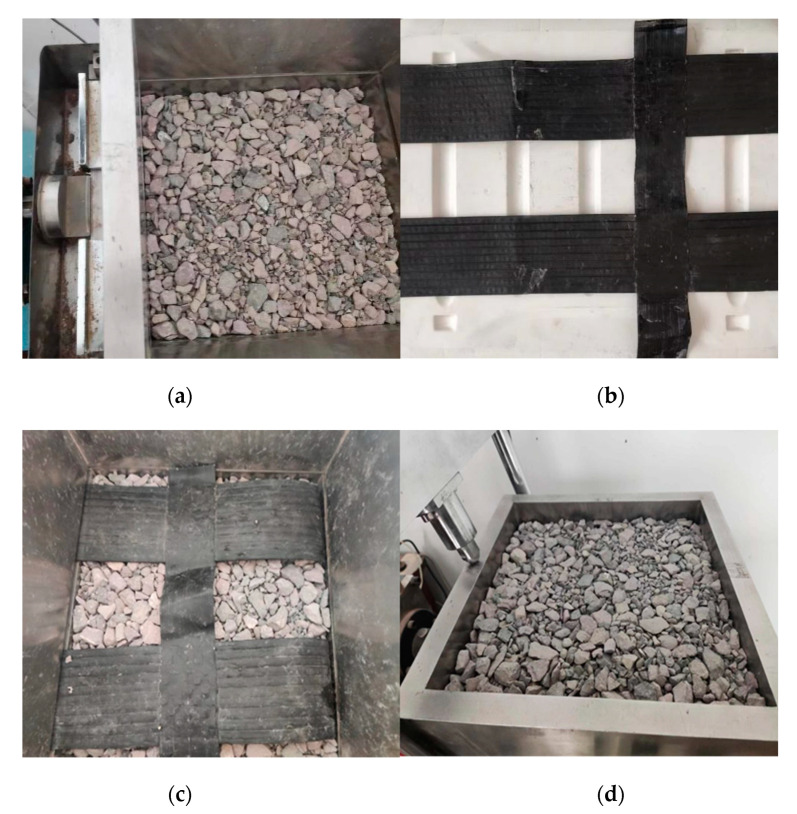
Sample preparation procedure. (**a**) Fill the lower shear box; (**b**) cut the geogrid; (**c**) lay the geogrid; (**d**) fill the upper shear box.

**Figure 7 materials-16-05858-f007:**
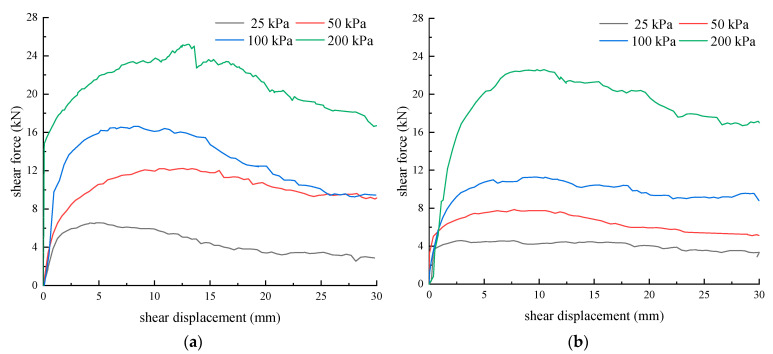
Shear force–shear displacement curve of crushed rock aggregates under different normal stresses. (**a**) Unreinforced; (**b**) PL geogrid-reinforced.

**Figure 8 materials-16-05858-f008:**
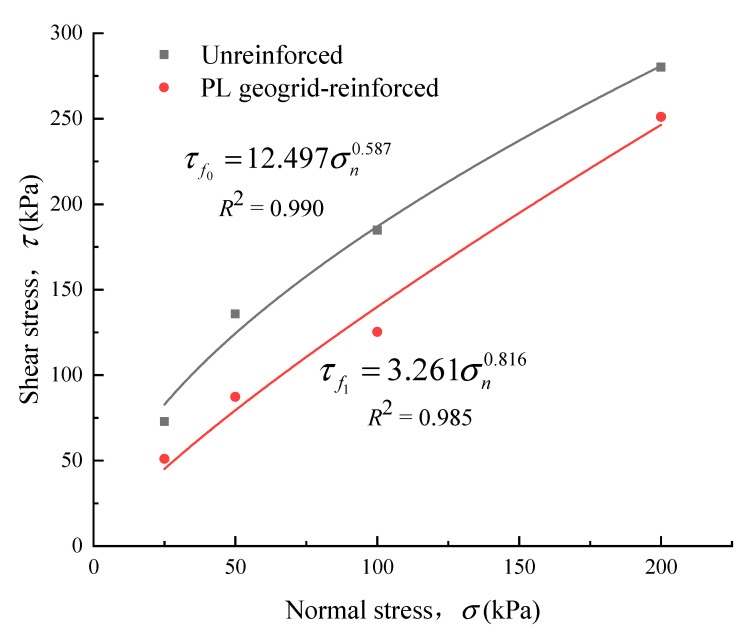
Shear strength envelope of the crushed rock aggregates.

**Figure 9 materials-16-05858-f009:**
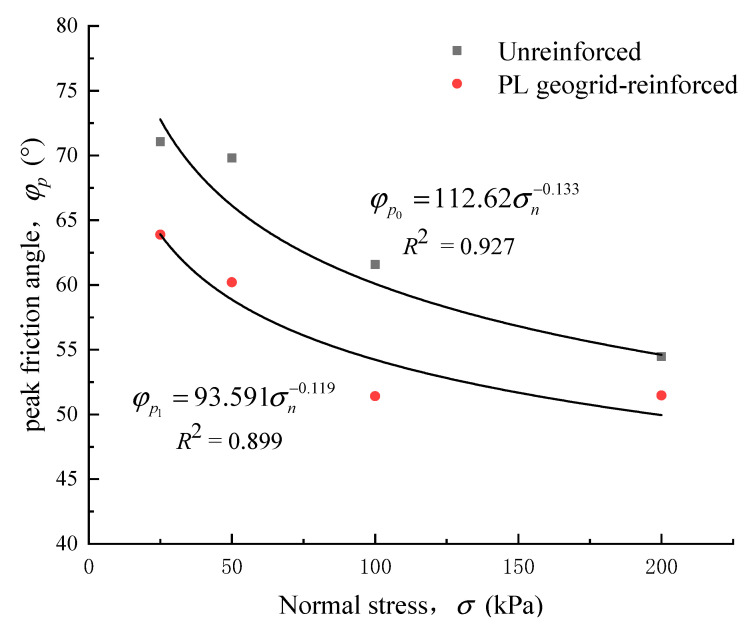
Relationship curve between peak friction angle and normal stress.

**Figure 10 materials-16-05858-f010:**
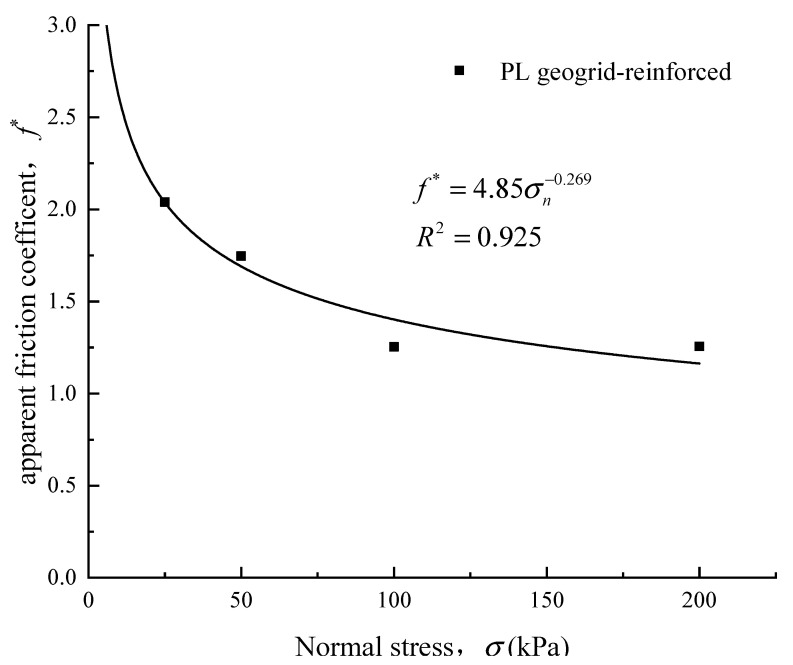
Relationship curve between apparent friction coefficient and vertical stress.

**Figure 11 materials-16-05858-f011:**
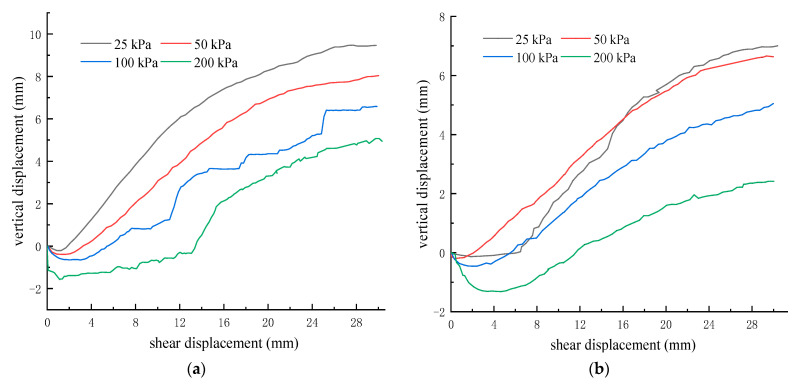
The vertical displacement-shear displacement relationship curves for unreinforced and PL geogrid − reinforced crushed rock aggregates. (**a**) Unreinforced; (**b**) PL geogrid − reinforced.

**Figure 12 materials-16-05858-f012:**
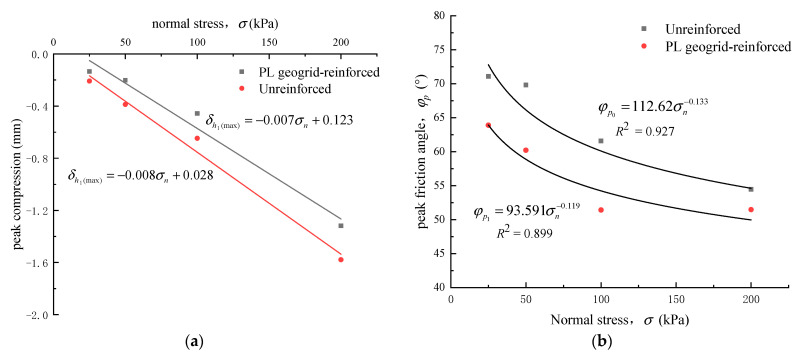
Relationship curves between the peak compression, peak dilation angle, and normal stress. (**a**) peak compression; (**b**) peak dilation angle.

**Figure 13 materials-16-05858-f013:**
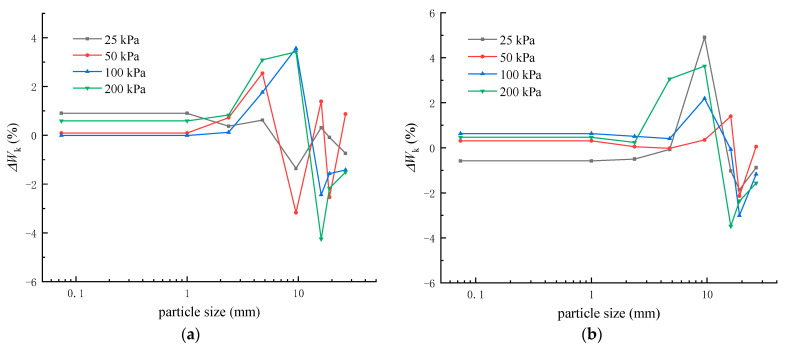
Variation in Aggregates Composition. (**a**) Unreinforced (**b**) PL geogrid-reinforced.

**Figure 14 materials-16-05858-f014:**
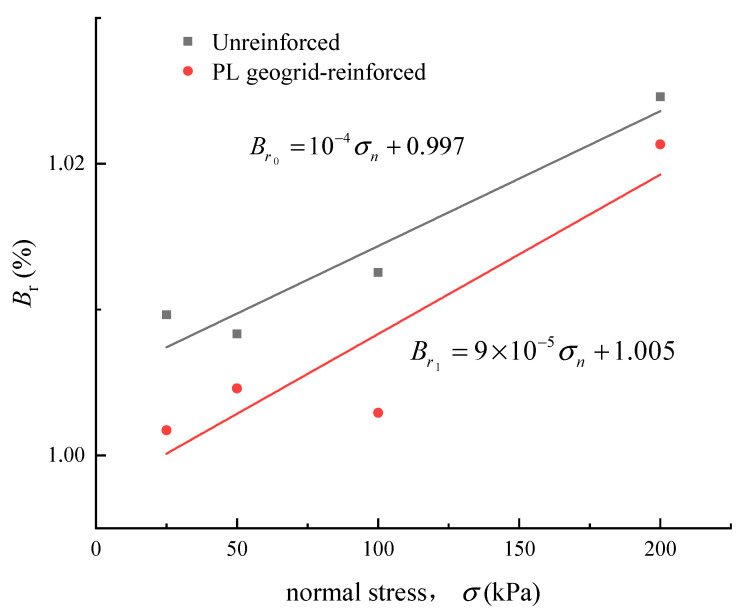
Relationships between *B*_r_ and normal stress.

**Table 1 materials-16-05858-t001:** Main performance indexes of large direct shear instruments.

Project	Performance
Load capacity	30 kN
Displacement rate range	0.003~3 mm/min
Horizontal range	50 mm
Vertical range	50 mm

**Table 2 materials-16-05858-t002:** Main characteristics of the geogrid.

Geogrid	Mass per Unit Area (g/m^3^)	Aperture Size (mm)	LD and TD Ribs Width (mm)	Ultimate Tensile Strength (kN/m)
ParaLink Geogrid	860	90 × 930	60.90	581

## Data Availability

Data is contained within the article.

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
