# Peer review of "Experimental Study on the Interface Characteristics of Reinforced Crushed Rock Cushion Layer Based on Direct Shear Tests"

_materials, 2023, doi:10.3390/ma16175858_

Round 1

Reviewer 1 Report

The article was carefully studied by the reviewer, and the following conclusions were made:

1. The decrease in shear strength with the addition of geotextile suggests that its characteristics are not properly selected. In section 3, there is no discussion of how the authors' results on the reduction in shear strength when using geotextiles are consistent with the results of other researchers.

2. The bibliography contains only 10 papers no older than 10 years, of which only 2 have been published in the last 5 years. This raises the question of the relevance of the work at the moment.

There are also less important remarks:

1. There is no explanation of what values are presented in Table 2. What is d10, d30, etc.?

2. The title of figure 9 needs to be edited so that it does not appear as a sentence.

3. The sentences in lines 399-400 and 406-407 are duplicated.

Proofreading is required. There are some grammatical errors and typos. For example, line 44: "base on" instead of "based on", line 79: "focuses" instead of "focuse", line 108: "apparents" instead of "apparatus", etc.

Reviewer 2 Report

The manuscript entitled “Experimental Study on the Interface Characteristics of Reinforced Crushed Rock Cushion Layer Based on Direct Shear Tests ” presents an interesting research related to the determination of friction parameters between geogrid and aggregate, important for engineering tasks. The article is of an engineering nature, the scientific and innovative aspects are weak in it! What is the hypothesis of the research conducted only on these selected materials?

In my opinion, the abstract should be more condensed and should present the most important outcomes and highlight the novelty.

The introduction should be enhanced by referring to recent studies covering the friction parameters, and the impact of the results on the stabilization of the ground and the stability conditions of the reinforced slopes.. 

The “Experimental Research” section should be clarified by adding the detailed description of the geogrid and agregate parameters, used in the tests. In this aspect, it is required to clarify the procedure of adopting materials parameters. Where are the mentioned geotechnical parameters required as the material input? I cannot get the idea of their use further in the test.

The description of the requirements for geogrids and aggregates and the criteria for their selection should be developed. Why were only these selected materials used in the research?

The Discussion is very weak. Hence, I suggest to enhance this chapter and refer to some more articles to compare main findings of the study and to formulate conclusions valid in a broader scale. The novelty and originality of performed research should be also strengthened. 

Please focus also on the technical aspect of the manuscript, especially check and verify the use of superscripts and subscripts in formulas and units as well as please avoid typos throughout the text. 

Reviewer 3 Report

Dear Authors,

The article should be reconstructed as it looks too much like a report and not a scientific article. It should also be noted that the research is stochastic in nature and should be repeated at least several times. Measurement uncertainties should also be specified. Therefore, please add to the article the number of times the research was repeated and the values of the measurement uncertainties associated with the research.

Moreover:

- equations prepared not in accordance with the requirements of the publisher

- no information on what the symbols in Table 2 mean

- the concept of layer 0 and 1 regarding the models under investigation is confusing. I propose model 0 and 1

- there is a difference in the description of equations (1), (2) and (3) and references to them, e.g. 3-1.

- l. 278 - 288 too many repetitions jp

- Fig. 11 lines are thinner than in other figures

- literature should be standardized according to the requirements of the publisher

Round 2

Reviewer 1 Report

All comments of the reviewer have been eliminated.

Author Response

Response to Reviewer #1

  1. The decrease in shear strength with the addition of geotextile suggests that its characteristics are not properly selected. In section 3, there is no discussion of how the authors' results on the reduction in shear strength when using geotextiles are consistent with the results of other researchers.

Answer:   Thanks for comments. Studies have shown that the length, width, thickness of ribs, grid node thickness, relative open area, and grid section of the geogrid all have an impact on the shear strength of the reinforced structure. The cross-sectional thickness of the ribs and nodes, as well as the relative open area of the geogrid, are positively correlated with the shear strength of the reinforced structure. On the other hand, the length and width of the ribs are negatively correlated with the shear strength of the reinforced structure. The interaction between geogrid and crushed rock aggregate mainly occurs through friction and interlocking effects.     

Consistent with the findings of Li and Zhao et al, the shear strength of PL geogrid-reinforced crushed rock aggregate is weakened compared to that of the unreinforced crushed rock aggregate. This is mainly attribute to the relatively small cross-sectional thickness of the PL geogrid-reinforced ribs and nodes, and the relatively large length and width of the ribs, as well as the smooth surface of the longitudinal ribs and fewer in quantity, the friction and interlocking forces at the interface between geogrid and  crushed rock aggregate are smaller compared to the bite force, friction force, and crushing force between the crushed rock particles. As a result, the shear strength of the PL geogrid-reinforced crushed rock aggregate is lower than that of unreinforced crushed rock aggregate.

In the newly developed subgrade structure with a synergistic interaction between the slope toe wall and the geogrid-reinforced cushion layer by our research team, the PL geogrid-reinforced is cast inside the concrete slope toe wall. The slope toe wall serves as an anchor segment for the reinforced cushion layer to compensate for the weak layer created by the geogrid in the crushed rock. The primary purpose of reinforcing the geogrid is to ensure the stability of the railway subgrade in the mining area and reduce uneven settlement deformation. When the geogrid achieves the desired tensile strength and effectively mitigates uneven settlement deformation, it remains highly applicable in this interlocking subgrade structure.

  1. The bibliography contains only 10 papers no older than 10 years, of which only 2 have been published in the last 5 years. This raises the question of the relevance of the work at the moment.

Answer:   Thank you for your comments. As a new type of geosynthetic material, geoglage has been widely used in various reinforcement projects, and the study of its interface characteristics in different reinforcement projects has always been a focus of attention. According to your suggestion, the relevant research literature has been added.

Our research group has developed a new subgrade structure based on the interlocking interaction between concrete sloped toe wall and mattress. In order to meet the need of stress transfer mechanism analysis of the new subgrade structure system, the paper conducted an experimental study on the interface shear behavior between crushed rock and the high-strength ParaLink Geogrid grille.

There are also less important remarks:

  1. There is no explanation of what values are presented in Table 2. What is d10, d30, etc.?

Answer:   Thanks for comments. In this study, there were some issues with the selection of material parameters in the experimental research. We have removed irrelevant parameters and clarified the procedure for adopting material parameters.

  1. The title of figure 9 needs to be edited so that it does not appear as a sentence.

Answer:   Thanks for comments. The title of  Figure 9 has been modified.

  1. The sentences in lines 399-400 and 406-407 are duplicated.

Answer:   Thanks for comments. Duplicate parts have been deleted.

Comments on the Quality of English Language

Proofreading is required. There are some grammatical errors and typos. For example, line 44: "base on" instead of "based on", line 79: "focuses" instead of "focuse", line 108: "apparents" instead of "apparatus", etc.

Answer:   Thanks for comments. Related grammar errors and typos have been corrected.

Reviewer 2 Report

The authors introduce significant changes and additions to the manuscript. It is now more understandable and the quality of the presented results has been improved. It is also still possible to supplement the literature review and discussion with reference to research conducted by other scientists. However, the article may be accepted for publication in its present form. Good luck!

Author Response

Dear reviewer, thank you very much for your comments on the revision of our manuscript. 

Reviewer 3 Report

Dear Authors,

Review in the attached file.

Author Response

Dear Reviewer:

 I really appreciate your detailed review comments, which are very helpful to improve the manuscript. Changes have been made to the previously submitted article.  All changes are highlighted in the text and  marked in blue.

Response to Reviewer #3

1、 after the sign equals, sometimes there is a space, and sometimes there is not

Answer: Thanks for comments, it has been modified. Changes are marked in blue

2、 abbreviation PL (PL geogrid-reinforced) has not been introduced

Answer: PL (PL geogrid-reinforced) has been introduced in Section2:

“The high-strength ParaLink ( the following text will be abbreviated as PL ) geogrid is used in the new subgrade structure. ”

3、 literature has not been corrected properly, e.g.:

Answer: The relevant literature has been corrected in accordance with your comments
